# MAPK Signaling Is Required for Generation of Tunneling Nanotube-Like Structures in Ovarian Cancer Cells

**DOI:** 10.3390/cancers13020274

**Published:** 2021-01-13

**Authors:** Jennifer M. Cole, Richard Dahl, Karen D. Cowden Dahl

**Affiliations:** 1Kabara Cancer Research Institute, Gundersen Medical Foundation, La Crosse, WI 54601, USA; jmcole@gundersenhealth.org; 2Department of Microbiology and Immunology, Indiana University School of Medicine, South Bend, IN 46617, USA; richdahl@iupui.edu

**Keywords:** ovarian cancer, tunneling nanotubes (TnTs), MAPK, macrophages, EGFR, tumor microenvironment

## Abstract

**Simple Summary:**

Ovarian cancer is the 5th leading cause of cancer death in US women, due to late diagnosis. The vast majority of patients with ovarian cancer have metastatic disease at diagnosis, leading to poor survival. As the tumor cells metastasize, they are influenced by other cells they encounter. In particular, we found that macrophages induce a mechanism of communication in tumor cells called tunneling nanotubes. These tunneling nanotubes allow cells to share molecules that promote metastasis. We found that macrophages send signals to the tumor cells in order to activate oncogenic MAPKinase signaling, which is required for tunneling nanotubes to form. Our new understanding of these events will enable us to devise ways to target tunneling nanotubes and limit tumor spread.

**Abstract:**

Ovarian cancer (OC) cells survive in the peritoneal cavity in a complex microenvironment composed of diverse cell types. The interaction between tumor cells and non-malignant cells is crucial to the success of the metastatic process. Macrophages activate pro-metastatic signaling pathways in ovarian cancer cells (OCCs), induce tumor angiogenesis, and orchestrate a tumor suppressive immune response by releasing anti-inflammatory cytokines. Understanding the interaction between immune cells and tumor cells will enhance our ability to combat tumor growth and dissemination. When co-cultured with OCCs, macrophages induce projections consistent with tunneling nanotubes (TnTs) to form between OCCs. TnTs mediate transfer of material between cells, thus promoting invasiveness, angiogenesis, proliferation, and/or therapy resistance. Macrophage induction of OCC TnTs occurs through a soluble mediator as macrophage-conditioned media potently induced TnT formation in OCCs. Additionally, EGFR-induced TnT formation in OCCs through MAPK signaling may occur. In particular, inhibition of ERK and RSK prevented EGFR-induced TnTs. TnT formation in response to macrophage-conditioned media or EGFR signaling required MAPK signaling. Collectively, these studies suggest that inhibition of ERK/RSK activity may dampen macrophage-OCC communication and be a promising therapeutic strategy.

## 1. Introduction

Ovarian cancer (OC) cells interact with non-malignant cells to facilitate survival, growth, metastasis, and immune evasion. Understanding the different approaches whereby tumor cells connect and transmit information to support cells in the tumor microenvironment (TME) will enhance our ability to combat tumor growth and dissemination. As ovarian cancer cells (OCCs) detach from the primary tumor and spread throughout the peritoneal cavity the OCCs interact with immune cells, mesothelium, fibroblasts, and omental adipocytes [1]. Just as we are no longer limited to communication via a house phone, tumor cells utilize different modes of communication in addition to canonical signal transduction cascades. By understanding the mechanisms of cellular communications, we can generate malware to disable tumor cell interactions and render the TME more susceptible to therapeutic interventions.

Non-traditional signaling mechanism are emerging as crucial modes of signal distribution. Exosomes and other extracellular vesicles contribute to metastasis by delivering metabolites, proteins, and RNA molecules that assist in fertilizing the pre-metastatic ovarian cancer niche in order to support tumor cell growth [2]. An additional form of communication for transferring components between cells is tunneling nanotubes (TnTs) [3,4]. TnTs are long, thin, actin-based, cytoplasmic extensions that form de novo and can serve as conduits for intercellular shuttling of cargo such as proteins, microRNA (miRNA), and mitochondria. TnTs transport oncogenic miRNAs between malignant cells and between malignant and stromal cells [5]. TnTs are fascinating structures in that different types or characteristics of TnTs are described. Thin TnTs only contain actin, yet thicker ones can also contain tubulin. These structures can be open ended or close ended, which changes how contents are shared between cells [6]. TnTs can also transmit information unidirectionally or bi-directionally [4,7,8]. Additionally, TnTs can be homotypic in that they connect similar cells (tumor cell to tumor cell) [7,9] or heterotypic meaning they connect different cell types (like macrophages to tumor cells) [10]. High-resolution microscopy and 3-dimensional reconstructions confirmed that TnTs are present in invasive malignancies, including human ovarian adenocarcinoma [5], providing evidence that TnTs are a characteristic of highly invasive cancers with metastatic potential, such as OC. Critically, the contribution of TnTs to tumor metastasis, immune evasion, and chemoresistance is not fully understood. Additionally, the conditions and pathways that lead TnT formation in cancer cells are not fully elucidated.

Human ovarian cancer is characterized by frequently metastasizing to the omentum [1]. OCCs detach from the primary tumor, adhere to the mesothelium, displace mesothelial cells, and invade into the layer of adipocytes that compose the omentum [1]. This metastatic process is conducted in the presence of other cells including fibroblasts and immune cells. Macrophages are one of the most abundant immune cell types in the TME which can be polarized to an M1 inflammatory phenotype that are considered classically activated and an M2 phenotype that promotes tumor cell metastasis, immune evasion, and chemoresistance [11]. High levels of M2 macrophages in ovarian tumors is indicative of a poor prognosis [11].

Understanding how the cells in the TME cooperate to drive tumorigenesis is a daunting task. In this study, we chose to isolate a few of these cell–cell interactions. In particular, we investigated the interactions between macrophages and ovarian tumor cells. By understanding the fundamentals of how cells in the TME communicate, we can move forward to identify ways to disable communication attempts and improve therapeutic outcomes.

## 2. Results

### 2.1. Ovarian Cancer Cell Interactions with Cells in the Tumor Microenvironment

We assessed the morphology of tumor cells in response to interactions with mesothelial cells, omental adipocytes, and immune cells employing a co-culture system to examine interactions within the TME.

### 2.2. Mesothelial Cell Culture with Ovarian Cancer Cells

For our studies we utilized three independent OCC lines. OVCA429 and OVCA433 cells are ovarian carcinoma derived cell lines, but since they have wild type TP53 they are not likely high grade serous ovarian cancer (HGSOC). Importantly, these cell lines are ideal OCCs for this study because they are normally non-invasive and display a typical epithelial morphology, but stimuli like epidermal growth factor (EGF) can induce an epithelial to mesenchymal transition, migration, and Matrigel invasion in these cells [12,13,14]. HGSOC is the most common subtype of OC and results in the most OC deaths. Therefore, we also used CAOV3 cells as they have genomic alternations consistent with HGSOC [15]. CAOV3 cells have an intermediate morphological phenotype; they are less epithelial than the OVCA429 and OVCA433 cells. First, we examined the relationship between OCCs and mesothelial cells. In order to examine distinct interactions, we co-cultured primary human mesothelial cells with OCCs (OVCA433 and OVCA429) as described [16]. Previously we found that SKOV3IP cells could clear mesothelial cells when plated on Collagen 1 using the established mesothelial clearance assay [16]. This method was first published using OVCA433 cells [17]. Mesothelial cells were plated at 100% confluency on 4 well chamber slides. After 24 h. OCCs were added to the mesothelial cultures. After 48 h, cultures were fixed and stained with Texas-Red Phalloidin (F-actin) and DAPI (nuclei). All co-culture experiments were conducted at least three times for each OCC line. The tumor cells retained typical epithelial clusters and maintain cortical actin which are indicative of an epithelial morphology (Figure 1A,B and Appendix A). The OCCs did not have membrane extensions or appear to become mesenchymal (Figure 1C,D). The mesothelial cells appear to surround the epithelial colonies which can be indicated by higher expression of keratin 8 which is a marker of mesothelial cells (Figure 1 and Appendix A) [18].

### 2.3. Omental Adipocytes Cultured with Ovarian Cancer Cells

We then explored the relationship between omental adipocytes and OCCs. Human omental pre-adipocytes were plated in a confluent layer on 4-well chamber slides. The pre-adipocytes were obtained from Zen Bio and differentiated according to the manufacturer’s instructions by the addition of adipocyte differentiation media. OVCA433 and OVCA429 cells were lentivirally transduced with CD133-GFP. We used the CD133-GFP since we and others have demonstrated that CD133 is localized to the plasma membrane and membrane extensions [16,19,20,21]. OVCA429-CD133-GFP (Figure 1E) cells or OVCA433-CD133-GFP cells (Figure 1F) were added to differentiated adipocyte cultures. Cultures were fixed and stained with Texas-Red Phalloidin and DAPI. Of note, the lipid droplets displayed autofluorescence (green). OCCs mostly retained their epithelial colony morphology but did appear more disorganized and have some membrane extensions (more than the mesothelial cell/OCC cultures) (Figure 1E,F). The OCCs also did not appear to displace the adipocytes, since particularly in Figure 1F, GFP expressing OVCA433-CD133 cells can be seen on top of lipid droplets present in the adipocytes. (Figure 1E,F). For the OCC-adipocyte co-cultures, two independent batches of adipocytes were used. Three 4 well-slides were prepared and used in co-culture for the first round of adipocyte differentiation and five slides were used for the second round of adipocyte differentiation. Half of the slides were used for OVCA429 co-cultures and half were used for OVCA433 co cultures. Further studies are needed to understand the consequences of OCC adhesion to adipocytes.

### 2.4. Macrophage Interactions with Ovarian Cancer Cells

#### 2.4.1. Macrophages Induce Tunneling Nanotubes in OCCs

Another critical component to metastatic success is the contributions from the immune system. OCCs interact with both innate and adaptive immune cells in order to metastasize and evade immune elimination [22]. Since, this is a complex process, we focused on the interaction between OCCs and macrophages. We differentiated the monocytic THP-1 cell line into macrophages using Phorbol 12-myristate 13-acetate (PMA). This method was demonstrated to yield adherent macrophages that are CD68 positive which we confirmed (Appendix A) [23]. We then added OCCs (CAOV3, OVCA429, and OVCA433) to these cultures and conducted immunofluorescence (IF) for the monocytic marker CD14 followed by co-staining with Texas Red Phalloidin and DAPI (Figure 2). CD14 was used to identify the immune derived cells. This IF was also conducted for THP-1 cells polarized to the M1 and M2 phenotypes, as discussed in the next section. All co-culture and conditioned media experiments were conducted at least three times for each OCC line. We found that co-cultures induced morphological changes to the OCCs including loss of epithelial morphology, elongated cells, and actin-based protrusions, and actin-based membrane bridges between cells some of which are consistent with TnTs (Figure 2, see arrows) [24,25]. Figure 3 (left column) shows typical morphology of untreated OCCs. There currently is not a definitive TnT marker, therefore we relied on published criteria to classify TnT-like structures. The following criteria was used to classify TnTs: (1) Thin actin based membrane projections greater than 30 µm, (2) lack of adherence to the substratum of culture plates, (3) visualization of TNTs passing over adherent cells; (4) TNTs connecting two cells; and (5) a narrow base at the site of extrusion from the membrane. These criteria are based on published criteria [24,25].

#### 2.4.2. M1 and M2 Polarized Macrophages Induce Tunneling Nanotubes in OCCs

Next we evaluated the impact of macrophage polarization on OCC morphology and TnT formation. We differentiated THP-1 cells as described and then polarized them to either the M1 or M2 phenotype [23]. RT-qPCR was conducted on RNA isolated from differentiated THP-1 cells or cells treated with M1 or M2 inducing cytokines to demonstrate that the polarization was successful. TNF mRNA expression was induced in M1 cells compared to unpolarized macrophages (differentiated THP-1 cells); and FN1 was increased in the M2 macrophages (Appendix A). THP-1 cells were plated on chamber slides for 24 h. After 24 h the cells were washed with PBS and media was changed to THP-1 media or THP-1 with M1 or M2 inducing cytokines for an additional 24 h. Then OCCs were added to M1 (Figure 2, middle column) and M2 (Figure 2, right column) macrophages 24 h after polarization. Co-cultures were stained for CD14 as an indicator of the THP-1 derived cells, Phalloidin, and DAPI to demonstrate the morphology and actin based structures in each cell type (Figure 2). Then to assess if soluble factors from macrophages are the instigators of TnT-like projections, we generated conditioned media (CM) from differentiated THP-1 cells, M1, and M2 polarized cells. The co-cultures also enabled us to better examine morphological changes seen in the OCCs. We added CM to each of the OC cell types for 48 h prior to phalloidin staining. Interestingly, we found that TnTs appeared to be generated in OCCs in response to all macrophage conditioned media (Figure 3 and Appendix A) and OCCs did not undergo a complete epithelial mesenchymal transition as E-cadherin localization was maintained (Appendix A). CM from all three macrophage types was sufficient to induce TnTs in OCCs. These data indicate that a secreted factor from macrophages is sufficient to alter the tumor cell morphology.

### 2.5. EGFR Induction of MAPK Signaling Results in TnTs

Our next task was to identify pathways that lead to formation of TnTs and similar membrane projections. We examined a number of growth factors to determine if they may promote TnT initiation including: EGF, HGF, NRG1, FGF2, FGF9, PDGF BB, and VEGF [26,27,28]. In the OCC lines, EGF and HGF significantly induced MAPK signaling in both the OVCA429 and OVCA433 cells (Appendix A). OVCA429, OVCA433, and CAOV3 cells plated on chamber slides and were serum starved for 24 h and then subsequently treated with growth factors for a final 24 h before being fixed for staining with Texas-Red Phalloidin and DAPI. In agreement with the lack of MAPK activation NRG1, FGF2, FGF9, PDGF-BB, and VEGF did not induce TnT formation (Appendix A) (*n* = 3). Intriguingly, the EGFR ligands, EGF and TGFα induced a mesenchymal morphology and loss of cell-cell junctions and TnTs in OCCs (Figure 4). This is consistent with EGFR activation inducing EMT, loss of E-Cadherin, and increased invasion [12]. We further dissected the signaling pathways that promote TnT formation (Figure 5). Two of the best studied pathways downstream of EGFR activation are MAPK and PI3K; therefore, we decided to investigate if inhibition of either of these pathways impact TnT induction. Chemical inhibitors were used to repress MAPK and PI3K signaling. OCCs were plated as described, serum starved, and treated with inhibitors of signal transduction and EGF or DMSO and EGF for 24 h. For quantification, images were collected at a 20× magnification, TnT length was measured (we only counted TnTs-like projections longer than 30 µm). We chose to count fields where there were between 100–300 cells per field to obtain consistent results. We noted that TnT formation varied by cell density. Sparsely populated areas yield more TnTs than dense epithelial sheets. This is why we tried to consistently only analyze cells in which between 100–300 cells were in a field of view. Inhibitors of MEK (U0126) and ERK (SCH772984) both reduced TnT formation (Figure 5A,B). RSK (SL0101) inhibition also decreased TnT numbers (Figure 5B). All experiments involving growth factors and inhibitors were conducted a minimum of 3 times for each cell line. Inhibition of PI3K signaling with wortmannin did not significantly decrease EGF-induced TnT formation.

### 2.6. ERK Activation Is Required for TnTs to Form in Response to Macrophage Signals

Since in the OCC TnT formation relied on MAPK downstream of EGFR signaling, we assessed if MAPK activation is necessary for macrophage induction of TnT-like structures. We cultured OCCs in the presence of conditioned media from differentiated THP-1, M1, and M2 macrophages. This was done in the presence and absence of the ERK inhibitor SCH772984. In agreement with our studies, TnTs (or long-membrane protrusions) seen in Figure 3 were inhibited (Figure 6) by SCH772984, and cells retained a more epithelial morphology compared to OVCA cells treated with macrophage conditioned media. These data indicate that ERK activation plays a pivotal role in the acquisition of TnT-like projections.

## 3. Discussion

Cellular communication is critical for metastatic success. In the ovarian cancer metastatic milieu OCCs come into contact with mesothelial cells, adipose tissue, endothelial cells, fibroblasts, and immune cells [1]. These cell types assist tumor cells in growth and metastatic colonization by providing growth factors and nutrients. Additionally, immune cells like M2 macrophages favor tumor metastasis [11]. Therapies that disrupt tumor cell interactions with cells in the TME may limit tumor spread.

Our initial goal was to evaluate how OCCs communicate with TME cells they will likely encounter during the metastatic process. OCCs displace the mesothelium in the peritoneal tissue to access ECM and invade deeper into adipose tissue [1]. We confirmed that OCCs effectively invade mesothelium [16], and we found that although the OCCs displace the mesothelial cells they retain an epithelial morphology. In agreement with studies by Dr. Stack, TnTs were not found in mesothelial-tumor cell co-cultures in the absence of external pressure. In their studies, it was found that compression increased TnT formation between peritoneal cells and tumor cells [29]. Ongoing studies suggest that mesothelial/OCC induce production of FGFs and EGF growth factors in mesothelial cells that may create a paracrine loop that enables tumor cell growth and invasion. Future studies will elaborate on the importance of mesothelial cell communication with OCCs.

Next, we explored what happens when the OCCs are exposed to omental adipose cells. In contrast to the OCC cultures with mesothelial cells, OCCs did not displace the adipocytes. In some cases (as can be seen in Figure 1F) GFP expressing OCCs were on top of adipocytes (evident by the fluorescent lipid droplets). The OCCs also underwent morphological changes including some disorganization of the epithelial morphology and some membrane protrusions. OCCs utilize adipocytes as an energy source [30]. Future efforts will be conducted to characterize the crosstalk between the OCCs and the adipocytes.

The most fascinating morphological changes we identified were cellular bridges between tumor cells in response to macrophages. These actin-based structures are consistent with TnTs. Yet the possibility remains that multiple types of membrane projections are induced by macrophages. Moreover, studying these structures and how they are formed may provide insight into the communication between macrophages and tumor cells. We found that a (not yet identified) soluble factor produced by macrophages (in the conditioned media) resulted in OCCs generating TnT-like structures. TnTs are important for cellular communication such as transmitting mitochondria, miRNA, and even oncogenes like KRAS [5,25,31] between cells. While we are not certain what information is being shared in response to macrophages, we did find that some of these structures contain mitochondria (Appendix A). The factor secreted by macrophages to induce TnTs did not induce a complete EMT phenotype in OCCs, yet EGFR induces both TnTs and EMT [10,12,13]. TnTs were formed without a loss of E-cadherin expression or a complete loss of cell–cell junctions. In agreement with this observation, EGFR inhibitors did not prevent TnT formation resulting from macrophage conditioned media (not shown). It was previously reported that differentiated THP-1 cells secrete EGF that can act on OCCs to influence proliferation and invasion [14]. In our hands the differentiated THP-1 cells increased TnT formation, but E-cadherin was retained at adherens junctions and Erlotinib and AG1478 did not prevent TnT formation (Appendix A). This suggests that another soluble signal in our THP-1 cultures promotes TnTs. Of note we did treat cells with IL-13 and IL-4 and LPS and IFN-**γ** (Appendix A). These factors did not induce TnT/membrane projection formation. Importantly, we made sure that all of our co-cultures and CM did not contain PMA (media on cells was changed after 24 h of PMA treatment) since PMA induces EGFR [32] (Appendix A). Therefore, an unidentified soluble factor is generated by macrophages that leads to TnT induction, but not EMT.

The field of TnTs is still maturing. There is not a definitive TnT marker and criteria vary somewhat between studies. Importantly functional criteria such as real time visualization that cargo is passed between cells also provides strong evidence of TnT identity. We conducted immunofluorescence on differentiated THP-1 co-cultures with OVCA429 cells for the mitochondrial protein TOM20 to show that a fraction of the TnT-like structures appears to have mitochondria (Appendix A). We tried to use stringent measures of TnT criteria to limit the chance that we were counting filipodia, tails of dividing cells, or other unidentified membrane protrusions. We did not count projections that were thin, as noted with an asterisk in Appendix A. We also did not count TnT-structures that were less than 30 um in length. Many of the TnTs we noted passed over adjacent cells before connecting with the target cell. Further evaluation with alternative microscopy approaches like higher resolution imaging or electron microscopy will strengthen the evidence that macrophages induce TnTs. Importantly, our work agrees with published studies that macrophages induce TnTs.

More work is needed to understand how macrophage-induced TnTs might influence immunity and metastasis. Although M2 polarization is often associated with tumor promotion, macrophage polarization did not impact TnT formation as naïve THP-1 macrophages and cells polarized to the M1 and M2 phenotypes all resulted in TnT formation. Further investigations will explore the functional impact of macrophage polarization of the tumor cells. Critically, we found that macrophages evoke MAPK signaling in the OCCs as the ERK inhibitor decreased TnTs. It was previously shown that EGFR activation by macrophages leads to TnTs [10]; we did not find EGFR activated in our studies. This may be a difference in the source of the macrophages. The previous study found that RAW/LR5 macrophages produced heterotypic TnTs with tumor cells and this required EGFR signaling. We typically observed that TnTs joined tumor cells and did not connect tumor cells to macrophages. However, we did find that RAW 264.7 conditioned media modestly induced TnTs in a mouse ovarian cancer cell line (ID8p53^−/−^) (Appendix A) [24]. It will be interesting to explore macrophage tumor interactions with different populations and sources of macrophages including primary peritoneal macrophages. Importantly, our studies agree with previous reports that demonstrate that EGFR is a major regulator of TnT formation, and we have further shown that ERK and RSK are effectors responsible for formation of structures consistent with TnTs that connect ovarian tumor cells. Further studies are necessary to identify the macrophage secreted factors that modulate TnT formation in OCCs.

Finally, we demonstrated that inhibition of RSK, a target of ERK, is sufficient to prevent TnT formation. RSK was demonstrated to be critical to both hematogenous and peritoneal spread of ovarian cancer [33]. RSK promotes metastasis by promoting seeding of metastatic cells [33]. Additionally, RSK regulates actin polymerization and cell motility [34]. In conclusion, RSK is a promising MAPK effector that could be crucial in ovarian cancer metastasis in part through regulation of TnT formation. Ongoing investigations will assess the impact of RSK on TnTs and how signals from the TME may promote metastasis via RSK.

## 4. Materials and Methods

### 4.1. Cell Culture

Cell lines were grown at 37 °C with 5% CO_2_. OVCA429 and OVCA433 cells (provided by Dr. Bast, MD Anderson Cancer Center, Houston, TX, USA) [35] were grown in Minimal Essential Medium (MEM) ThermoFisher (Waltham, MA, USA); The media was supplemented with 10% fetal bovine serum (FBS) (Peak Serum, CO, USA), 0.1 mM L-glutamine, 1 mM sodium pyruvate, 50 U/mL penicillin, and 50 μg/mL streptomycin (all from ThermoFisher (Waltham, MA, USA)). CAOV3 and RAW 264.7 cells were purchased from ATCC (Manassas, VA, USA) and maintained in DMEM Media was supplemented with 10% FBS (Peak serum, CO, USA), 0.1 mM L-glutamine, 1 mM sodium pyruvate, 50 U/mL penicillin, and 50 μg/mL streptomycin. Human Mesothelial cells were a kind gift from (Dr. Hilary Kenny, University of Chicago) and grown in RPMI-1640 with 20% FBS, 0.1 mM L-glutamine, 50 U/mL penicillin, and 50 μg/mL streptomycin, MEM non-essential amino acids and MEM vitamins. ID8p53^−/−^ (ID8 clone F3) murine ovarian cancer cell lines were provided by Iain A. McNeish (University of Glasgow) [36]. Cells were cultured in DMEM, 4% FBS, 50 U/mL penicillin, 50 µg/mL streptomycin, and 1× ITS (5 µg/mL insulin, 5 µg/mL transferrin and 5 ng/mL sodium selenite). All tissue culture additives were purchased from ThermoFisher (Waltham, MA, USA). 75,000 human mesothelial cells were plated on each well of a 4 well chamber slide that was coated with Collagen 1. After 24 h, 10,000 OVCA429, or OVCA433 cells were added to mesothelial cell cultures. Twenty four hours later cells were fixed for fluorescence. Human omental pre-adipocytes were purchased from Zen Bio (Research Triangle Park, NC, USA). They were cultured/differentiated per Zen Bio’s instructions (https://www.zen-bio.com/pdf/ZBM0022.pdf). Upon receipt the pre-adipocytes were immediately thawed and plated confluently (70,000 cells/well) onto 4 well chamber slides. After 24 h pre-adipocytes were differentiated by the addition of Omental Differentiation Medium for 7 days. After 7 days Omental Adipocyte Medium was added for an additional 7 days. At that time 16,000 OVCA429-CD133-GFP or OVCA433-CD133-GFP was added to the adipocytes. After 5 days adipocyte/OCC cultures were fixed for immunofluorescence (IF). THP-1 monocytes (ATCC, Manassas, VA, USA) were cultured in 10% FBS in RPMI-1640 with 50 nM betamercaptoethanol. To differentiate THP-1 cells to macrophages, THP-1 cells were plated at 250,000 cells/mL and then treated with 150 mM PMA (Sigma-Aldrich, St. Louis, MO, USA) for 24 h [23]. After 24 h, cells were rinsed with PBS and THP-1 media (without PMA or cells are polarized). To polarize macrophages to macrophages to the M1 phenotype cells were treated for 24 h with 100 ng/mL LPS and 20 ng/mL IFN-**γ** for 24 h. M2 polarization was accomplished by treating cells with 20 ng/mL IL13 and 20 ng/mL IL4 for 24 h. To validate successful polarization differentiated RNA was collected using Trizol from THP-1, M1, or M2 polarized cells after 24 h post-polarization. For co-cultures, 12,000 CAOV3, OVCA429, or OVCA433 cells were added to THP-1 cultures (125,000 cells/well). After 48 h cells were fixed with formaldehyde for fluorescence analysis.

OVCA433 and OVCA429 cells were lentivirally transduced with Green Fluorescent Protein (GFP, Genecopia, Rockville, MD, USA), and PROM1-GFP (Genecopia, Rockville, MD, USA).

For growth factor and inhibitor studies, 10,000 CAOV3 cells or 7000 OVCA429 or OVCA433 cells were plated on 4 well chamber slides. Twenty four hours after plating, OCCs were serum starved for 24 h and then treated with growth factors, DMSO and EGF, or EGF and indicated chemical inhibitors for an additional 24 h. After treatment cells were fixed and stained for actin and nuclei. To investigate pathways that contribute to TnTs the following growth factors were used 20 nM EGF, 50 ng/mL TGFα, 20 ng/mL NRG1, 4 ng/uL PDGF BB, 50 ng/mL FGF2, 20 ng/mL FGF9, 20 ng/mL VEGF, and 50 ng/mL HGF. FGFs were added with 1 µg/µL heparin.

Additionally, the following chemical inhibitors of signal transductions were used: 2 µM SCH772984 (ERK inhibitor), 200 µM SL0101-1 (RSK inhibitor), 5 nM Wortmannin (PI3K inhibitor), and 10 µM U0126 (MEK inhibitor). Chemicals were obtained from Sigma, St. Louis, MO, USA. 

### 4.2. Western Blots

OVCA429 and OVCA 433 cells were serum starved for 24 h. Cells were then treated with growth factors (as indicated above) for 30 min. Whole-cell lysates were obtained by lysing OVCA429 and OVCA433 cells in RIPA (50 mM Tris pH7.5, 150 mM NaCl, 1% NP-40, 0.5% EDTA, and 1× Halt Protease Inhibitor Cocktail (Pierce, Rockford, IL, USA)). Protein concentration was determined using BCA assay according to standard protocol (Pierce, Rockford, IL, USA). Proteins were detected using the following antibodies: anti-phospho-p44/42 and anti-ERK(Rabbit polyclonal, Cell Signaling Technology, Danvers, MA, USA), and followed by a secondary anti-rabbit HRP-conjugated antibody (Cell Signaling Technology, Danvers, MA, USA). Imaging was conducted using a Bio-Rad ChemiDoc Touch Imaging System, running Imager Lab Software (Hercules, CA, USA). See Appendix A for full blots and densitometry.

### 4.3. Fluorescence

All fluorescent experiments were conducted a minimum of 3 times. Images depict representative images.

To examine morphology of OCC/human primary mesothelial, omental adipocyte, or macrophage co-cultures, cells were fixed with 3.7% formaldehyde in PBS. Cells were then permeabilized with triton and blocked with 3% bovine serum albumin (BSA) in Tris-buffered saline with 0.1% tween (TBST). Actin was stained with Texas Red-Phalloidin (made according to Invitrogen’s directions) (Invitrogen, Carlsbad, CA, USA) in a 1:2000 dilution in 3% BSA/TBST blocking buffer for 1 h. Slides were washed and mounted using slides; they were all mounted with Fluoromount G with DAPI (ThermoFisher (Waltham, MA, USA).

Additionally, some OCC/mesothelial co cultures were stained with keratin 8 (Keratin 8/18 mAB (4546 T) from Cell Signaling, Davers, MA, USA) as a marker of mesothelial cells [18].

For immunofluorescence (IF) on THP-1/OCC co-cultures, cells were fixed with 3.7% formaldehyde in PBS. Cells are then permeabilized with triton and blocked with 3% bovine serum albumin (BSA) in Tris-buffered saline with 0.1% tween (TBST). Macrophages were identified with anti-CD14 staining (Biolegend, San Diego, CA, USA) followed by anti-mouse Alexa Fluor488 secondary (ThermoFisher). To demonstrate that the THP-1 cells were properly differentiated, IF was conducted for CD68 on adherent THP-1 differentiated as described (anti-CD68 #333802, Biolegend, San Diego, CA, USA). Additional IF was conducted on OVCA429 and OVCA433 cells treated with CM for E-cadherin (mAB 1416, Abcam, Cambridge, MA, USA). Slides were mounted with Fluoromount G with DAPI (ThermoFisher).

Additionally, OCCs were treated for 48 h with conditioned media (CM) from differentiated and M1 and M2 polarized THP-1 cells in the absence of presence of 2 uM SCH772984 for 24 h. After treatment cells were fixed and stained for Texas Red-Phalloidin (Invitrogen, Carlsbad, CA, USA) and DAPI as described.

For growth factor and inhibitor studies, OCCs were treated as stained for actin/nuclei as described above.

To determine if mitochondria were present in membrane projection, (IF) on THP-1/OVCA429 co-cultures, cells were fixed with 3.7% formaldehyde in PBS. Cells are then permeabilized with triton and blocked with 3% BSA in TBST. IF was conducted for TOM20 (#sc-11415, Santa Cruz Biotechnology, Santa Cruz, CA, USA) and also stained for F-actin using Texas Red-Phalloidin (Invitrogen, Carlsbad, CA, USA).

### 4.4. TnT-Like Projection Identification and Quantitation

There currently is not a TnT marker, therefore we relied on published criteria to classify TnT-like structures. The following criteria was used to classify TnTs: (1) thin actin based membrane projections of greater than 30 µm, (2) lack of adherence to the substratum of culture plates, (3) visualization of TNTs passing over adherent cells; (4) TNTs connecting two cells; and (5) a narrow base at the site of extrusion from the membrane. These criteria are based on published criteria [24,25]. For quantification, images were collected at a 20 µ magnification, TnT length was measured (we only counted TnTs longer than 30 µm), and optimally at least 100–300 cells per field were counted. We noted that TnT formation varied by cell density. Sparsely populated areas yield more TnTs than dense epithelial sheets. This is why we tried to consistently only analyze cells in which between 100 and 300 cells were in a field of view. Each experiment was performed at least 3 times. A minimum of 3 fields of view were collected for each condition for each experiment. Therefore, we had 3 biological replicates for each experiment. The number of TnTs was averaged for the 3 independent fields and the error bars are the SEM between independent experiments. We reported the TnT quantitation as the average percentage of cells with TnTs. Of note some cells had multiple TnTs.

### 4.5. Microscopy

Images were collected using an EVOS M5000 (Invitrogen, Carlsbad, CA, USA) at 60×, 40×, and 20× magnifications. For quantification, images were collected at a 20× magnification, TnT length was measured (we only counted TnTs longer than 30 µm), and optimally at least 100–300 cells per field were counted.

#### RT-qPCR

RNA from differentiated THP-1 cells and differentiated THP-1 cells polarized to M1 or M2 was isolated via Trizol method (Thermo Fisher, Waltham, MA, USA). Next cDNA was generated from 2 µg of RNA using the High Capacity cDNA Reverse Transcription Kit (Life Technologies, Waltham, MA, USA) as directed. A reverse transcribed quantitative polymerase chain reaction (RT-qPCR) RT-qPCR was conducted for TNF (M1 polarization) and FN1 (M2). Reactions were run either using IDT Prime Time master mix (Coralville, IA, USA) or Sso Fast.

EvaGreen Supermix (Bio-Rad). All gene expression primer sets were obtained from Integrated DNA Technologies (Coralville, IA, USA) (Table 1). RT-qPCR reactions were run in triplicate and normalized to expression of GAPDH. RNA was isolated from 3 separate differentiations/polarizations of macrophages.

### 4.6. Statistics

Statistics were conducted on RT-qPCR data that was conducted on 3 independent experiments using Student’s t-tests on Prism, (GraphPad, San Diego, CA, USA). Statistical significance was assigned to comparisons with a *p*-value of 0.05 or lower. Statistical analysis on TnT quantitation (from 3 separate experiments) was conducted on GraphPad using 1-way ANOVA.

## 5. Conclusions

Finally, these studies provide some critical insight into communication between tumor cells and macrophages. We show that macrophages induce formation of TnT-like structures in tumor cells through MAPK signaling. TnTs are present in metastatic ovarian cancer [5,7,25], therefore, macrophages may encourage metastatic spread by activation of MAPK signaling in OCCs. Inhibition of TnT formation through ERK or RSK inhibitors may have the added benefit of limiting TnT production and metastasis.

## Figures and Tables

**Figure 1 cancers-13-00274-f001:**
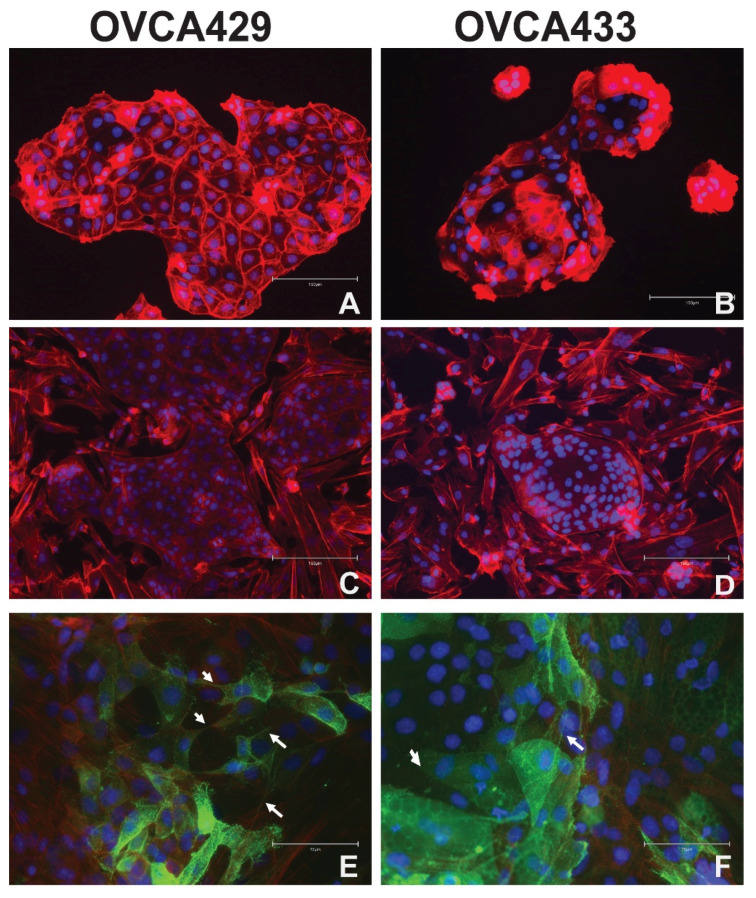
Morphology of ovarian cancer cells when co-cultured with mesothelial cells and omental adipocytes. Panels A and B depict the normal cobblestone epithelial morphology of OVCA429 (**A**) and OVCA 433 (**B**) cells when stained with Texas Red Phalloidin for F-actin (and DAPI for nuclei). OVCA429 (**C**) and OVCA433 (**D**) cells were cultured on top of a confluent layer of primary human mesothelial cells and stained with Texas Red phalloidin after 24 h. OVCA429-CD133-GFP (**E**) and OVCA433 CD133-GFP (**F**) cells were cultured on top of confluent differentiated omental adipocytes for 5 days. 40X original magnification. Scale bar = 75 µm. Green = GFP/adipocytes. Arrow = tumor cell projections.

**Figure 2 cancers-13-00274-f002:**
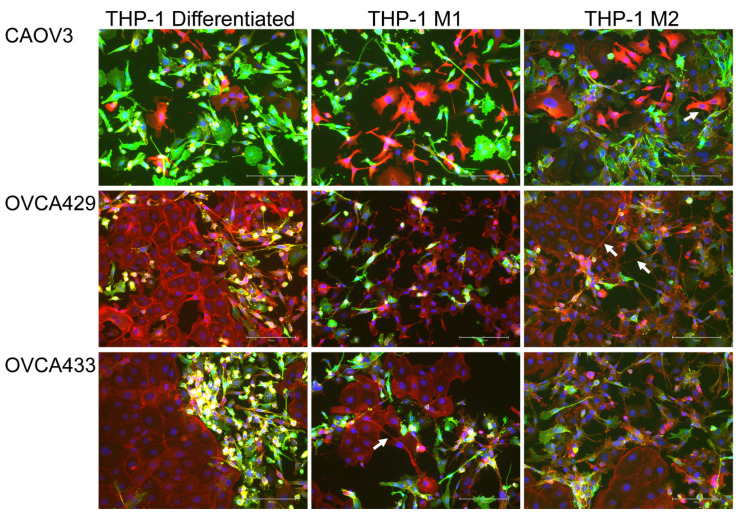
Macrophages alter cellular morphology of ovarian cancer cells during co-culture. THP-1 cells were differentiated and/or differentiated followed by subsequent polarization to M1 and M2 macrophages. After differentiation/polarization CAOV3, OVCA429, and OVCA433 cells were cultured with differentiated THP-1, M1, and M2 polarized macrophages for 48 h. Cultures were stained for CD14 to denote THP-1 derived cells (green), Texas Red Phalloidin as a marker of F-actin (red), and DAPI (blue). 20× original magnification. Scale bar = 150 µm. Arrow = denotes membrane projections produced between tumor cells.

**Figure 3 cancers-13-00274-f003:**
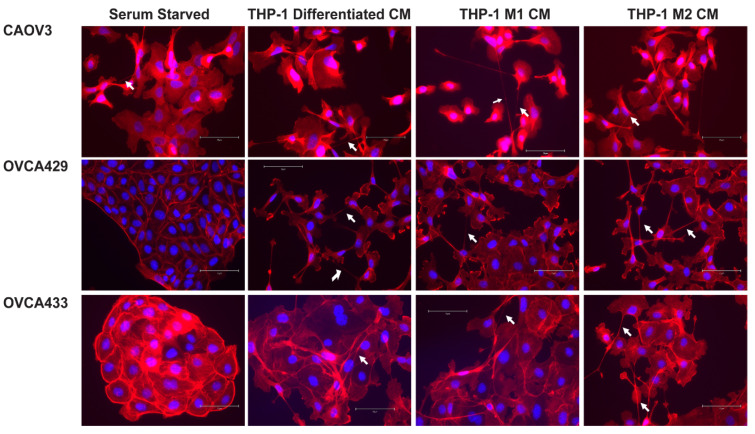
Macrophage conditioned media induces tunneling nanotubes (TnT)-like projections in ovarian cancer cells. CAOV3, OVCA429, and OVCA433 cells were serum starved for 24 h and left untreated or treated with conditioned media from differentiated THP-1, M1, and M2 macrophages for an additional 24 h. Cells were stained for Texas Red phalloidin (F-actin) and DAPI (blue). 20X original magnification. Scale bar = 150 µm. Arrow = long thin actin-based structures > 30 nM (TnT).

**Figure 4 cancers-13-00274-f004:**
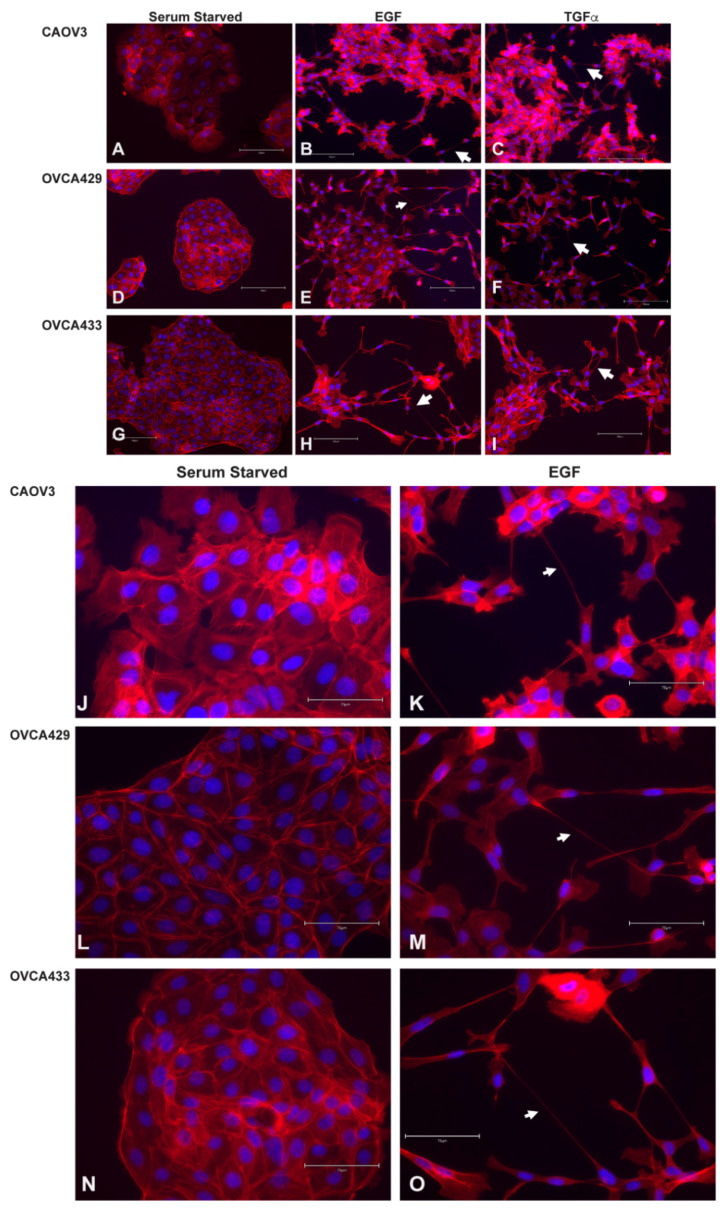
EGF and TGFα induce TnTs in ovarian cancer cell lines. CAOV3 (**A**–**C**,**J**,**K**), OVCA429 (**D**–**F**,**L**,**M**) and OVCA433 (**G**–**I**,**N**,**O**) cells were serum starved for 24 h and then treated with 20 nM EGF for 24 h or 50 ng/mL TGFα for 24 h. Cells were stained with Texas Red Phalloidin to visualize F-actin based structures and DAPI (blue). Arrows denote TnTs. (**A**–**I**) are 20X original magnification and (**J**–**O**) are 40X original magnification.

**Figure 5 cancers-13-00274-f005:**
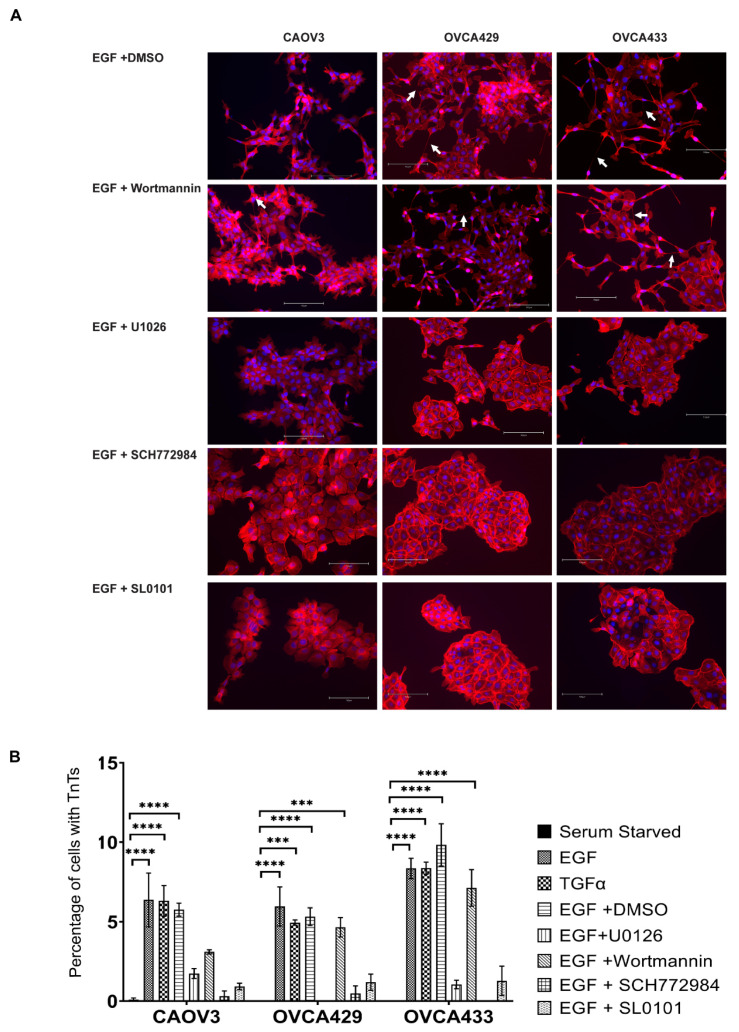
TnT formation requires MAPK activity. To determine if MAPK or PI3K signaling pathways contributed to TnT formation down stream of EGFR activity OCCs were treated with EGF and pathway inhibitors. (**A**) CAOV3, OVCA429, and OVCA433 cells were serum starved for 24 h and then treated for an additional 24 h with 20 nM EGF + DMSO, EGF + Wortmannin (PI3K inhibitor), EGF + U0126 (MEK inhibitor), EGF + SCH772984 (ERK inhibitor), or EGF + SL0101 (RSK inhibitor). Cells were stained with Texas Red Phalloidin to visualize F-actin based structures and DAPI (blue). Each experiment was performed in triplicate and a minimum of 3 fields of cells were analyzed for each treatment. (**B**) Total cell numbers and number of TnTs were counted for each field. Counts were averaged and error bars denote SEM between at least 3 biological replicates. For statistics, comparisons are between the EGF + DMSO and EGF + treatment. Arrow = TnT. *** *p* < 0.001, **** *p* < 0.0001.

**Figure 6 cancers-13-00274-f006:**
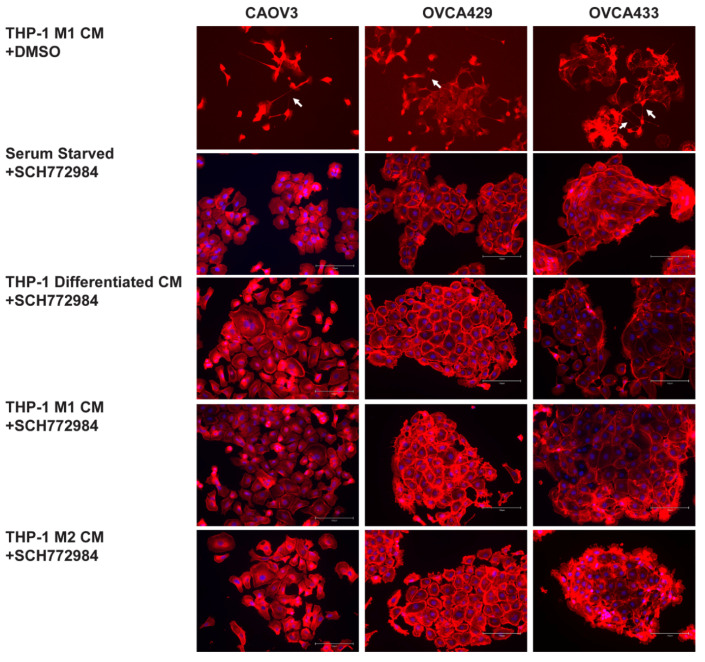
Membrane protrusions induction by macrophage conditioned media requires ERK activity. To assess if inhibition of ERK activity can prevent TnT formation in OCCs treated with macrophages, conditioned media (CM), CAOV3, OVCA429, and OVCA433 cells were serum starved for 24 h and then treated with conditioned media from differentiated THP-1, M1, and M2 macrophages in the presence of the ERK inhibitor SCH772984 for an additional 24 h. The top row depicts OCCs treated with M1 CM and DMASO, which results in TnTs and altered morphology (as seen in Figure 3). Cells were stained with Texas Red Phalloidin to visualize F-actin based structures and DAPI (blue). Images are representative of 3 independent experiments. 20X original magnification. Scale bar = 150 µm.

**Table 1 cancers-13-00274-t001:** Primers.

NCBI Gene Symbol	IDT Assay ID	Reference Sequence Number	Location
FN1	Hs.PT.58.40005963	NM_054034(6)	exon 3–4
GAPDH	Hs.PT.39a.22214836	NM_002046(1)	exon 2–3
TNF	Hs.PT.58.45380900	NM_000594(1)	exon 1b–4a

## Data Availability

Data is contained within the article and Appendix A.

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
