# Peer review of "MAPK Signaling Is Required for Generation of Tunneling Nanotube-Like Structures in Ovarian Cancer Cells"

_cancers, 2021, doi:10.3390/cancers13020274_

Round 1

Reviewer 1 Report

Overall, the manuscript is interesting but it is mostly a descriptive study. Other studies have shown that mesothelial cells and macrophages interact with ovarian cancer cells (OCCs) via tunneling nanotubes (TnTs). Yet, this is the first study to demonstrate that macrophages can induce OCCs to produce TnTs and mesothelial cells change OCC morphology. Macrophages or macrophage conditioned media can induce the formation of TnTs as well as purified EGF or TGF α. It is suggested that EGFR signaling leading to MAPK activation is required for OCC TnT formation. However, in data not shown EGFR signaling and therefore EGF is not the factor responsible for OCC TnTs via macrophage conditioned media. Macrophage polarization does not apparently alter the ability of macrophages to induce OCC TnTs. Higher resolution images, inclusion of data not shown and quantitation of the different effects (only figure 4 has statistical analysis) are needed to support the conclusions of this study. Also, the study could be improved with some mechanistic insight into the possible role of MAPK in TnT formation.

Specific comments

  1. Were the qRT-PCR arrays that indicated the changes in OCCs by co-culture with the mesothelial cell line only performed once? If so, then this data should be confirmed and potentially protein level should be assessed by western blot or flow. However, it is not clear how this data is relevant to the rest of the study.
  2. The displacement of the mesothelial cells by the OCCs is not clear in the images in Figure 1. Is there a way to measure the displacement? Also, the retention of tumor cell morphology is not apparent from the images. Perhaps the cell types can be differentially labeled and/or images of the tumor and mesothelial cells alone would be helpful.
  3. Please include a reference for the differentiation of adipocytes (manufacturer’s directions?).
  4. It is not clear what is meant by the extensive interaction of OCCs with adipocytes. Comparison to OCCs alone would be helpful. Why were the cells treated with the microtubule inhibitor paclitaxel especially if tunneling nanotubes are actin dependent? Could these “interactions” be quantified in some way?
  5. What was the ratio used in the cocultures with macrophages? The images in 2A show various numbers of macrophages (CD14). How does the right column in Figure 2B represent typical OCC morphology? Wouldn’t 2B left column of starved OCC cells alone be a more accurate representation of OCC morphology?
  6. It is unclear how large is the response of OCCs to macrophages? What fraction of the OCCs contain tunneling nanotubes in the various conditions? Some of the images are of too low magnification to clearly see cell morphology. Also, some of the structures indicated to be nanotubes appear to be too thick to be “nanotubes”.  Some of the long structures that don’t connect two cells together could be migratory tails. Is the EGF or TFGα inducing cell migration/ scattering or breaking down cell-cell junctions?
  7. EGF and TGF induce multiple signaling pathways leading to tunneling nanotube formation. Why was MAPK signaling examined? While nanotube formation is quantified in Fig. 4B it is not clear what ave%TnT represent. Does it represent number of cells with TnTs? The text said that 100-300 cells/field. Are the error bars the SD or SEM of independent experiments or fields?
  8. It is not clear how MAPK signaling is regulating TnT formation? Is this through gene expression or some other signaling pathway? How does PI 3-kinase and Akt signaling relate to MAPK activity?
  9. Control images of cells treated with CM with DMSO should be shown in figure 5 as a positive control.

A reference should be included for the Stack study

Author Response

We appreciate the thorough critique provided by this reviewer. The comments and suggestions have been taken into careful consideration and the manuscript including the figures have been extensively revised. These suggestions greatly enhance the paper and the findings that are reported. Please note responses in bold.

Overall, the manuscript is interesting but it is mostly a descriptive study. Other studies have shown that mesothelial cells and macrophages interact with ovarian cancer cells (OCCs) via tunneling nanotubes (TnTs). Yet, this is the first study to demonstrate that macrophages can induce OCCs to produce TnTs and mesothelial cells change OCC morphology. Macrophages or macrophage conditioned media can induce the formation of TnTs as well as purified EGF or TGF α. It is suggested that EGFR signaling leading to MAPK activation is required for OCC TnT formation. However, in data not shown EGFR signaling and therefore EGF is not the factor responsible for OCC TnTs via macrophage conditioned media. Macrophage polarization does not apparently alter the ability of macrophages to induce OCC TnTs. Higher resolution images, inclusion of data not shown and quantitation of the different effects (only figure 4 has statistical analysis) are needed to support the conclusions of this study. Also, the study could be improved with some mechanistic insight into the possible role of MAPK in TnT formation.

Thank you for the extensive comments. Yes, we have shown both that EGFR induces TnTs in OCCs and that an unidentified soluble factor from the macrophages also induce TnTs. Importantly, they both lead to MAPK activation. For the mechanism we were able to demonstrate that inhibition of ERK prevents TnT from being formed in response to EGFR or macrophages. We also demonstrated that inhibition of the ERK target RSK is also a critical component to TnT generation. Our current hypothesis is that RSK targets proteins that lead to TnTs. RSK has been demonstrated to be involved in ovarian cancer invasion and metastasis[1]. RSK does this by regulating transcription, integrin activity, and actin remodeling[2]. We have added larger figures, additional data, and quantitation to support our findings.

Specific comments

  1. Were the qRT-PCR arrays that indicated the changes in OCCs by co-culture with the mesothelial cell line only performed once? If so, then this data should be confirmed and potentially protein level should be assessed by western blot or flow. However, it is not clear how this data is relevant to the rest of the study.

Yes, the qRT-PCR arrays were performed once. Unfortunately, due to the recovery process from the FACSing, not enough cells were recovered for protein analysis. We have decided to exclude this data to focus on the macrophage study.

  1. The displacement of the mesothelial cells by the OCCs is not clear in the images in Figure 1. Is there a way to measure the displacement? Also, the retention of tumor cell morphology is not apparent from the images. Perhaps the cell types can be differentially labeled and/or images of the tumor and mesothelial cells alone would be helpful.

We appreciate the comment. Yes, displacement of the mesothelial cells by the ovarian cancer cells can be measured [3,4]. Dr. Brugge’s lab pioneered a mesothelial clearance assay using OVCA433 cells. We previously also published mesothelial clearance of SKOV3IP cells using this method[5]. We did not quantitate displacement in this current publication as we focused on morphology. In our previously published results, we used ovarian cancer cell labeled with RFP and mesothelial cells labeled with GFP (Roy et al) to show and quantitate that OCCs displace mesothelial cells to contact collagen I. However, we added the supplemental image showing that mesothelial cells were stained with cytokeratin 8 surround a group of tumor cells. Cytokeratin 8 has been demonstrated to be a mesothelial marker[6]

  1. Please include a reference for the differentiation of adipocytes (manufacturer’s directions?).

We apologize for the oversight. The manufacturer’s directions on adipocyte differentiation have been added.

  1. It is not clear what is meant by the extensive interaction of OCCs with adipocytes. Comparison to OCCs alone would be helpful. Why were the cells treated with the microtubule inhibitor paclitaxel especially if tunneling nanotubes are actin dependent? Could these “interactions” be quantified in some way?

We have revised the text about regarding the OCCs and adipocytes to be more clear. We apologize for not clearly stating the rationale for these experiments. These points are clarified in the test. OCCs alone (Fig. 1) form epithelial sheets. When OCCs were co-cultured with the omental adipocytes, the OCCs had changes in morphology including some membrane extensions some disorganization of the epithelial sheets. We investigated the role of paclitaxel for a couple reasons. First, carboplatin and paclitaxel are the standard of care chemotherapy for patients with ovarian cancer. Additionally, we previously found that low dose (non-cytotoxic) (7.5nM) paclitaxel inhibits the acquisition of TnT-like extensions in cells treated with EGF. We did not end up including this data because we have not yet identified the mechanism. A previous report suggested that paclitaxel inhibits EGFR signaling [7]. But we could not replicate those findings and found no change in EGFR activation in cells treated with paclitaxel. Therefore, in order to better streamline the narrative, we will remove this data and focus on the monocyte/macrophage story.

  1. What was the ratio used in the cocultures with macrophages? The images in 2A show various numbers of macrophages (CD14). How does the right column in Figure 2B represent typical OCC morphology? Wouldn’t 2B left column of starved OCC cells alone be a more accurate representation of OCC morphology?

Thank you for the thought provoking question. We did not test different ratios of macrophages to tumor cells. With the mesothelial cells and the adipocytes, tumor cells were plated on a confluent layer of cells (either mesothelial or adipocyte). We did not plate confluent THP-1 cells. However we wanted the THP-1 cells to be dense enough that there was a likelihood that they would come into contact with tumor cells. The THP-1 cells are much smaller than the tumor cells. Therefore we plated 125,000 THP-1 cells per well of a 4 well chamber slide and made certain that the cells were even dispersed evenly on the slide. The number of tumor cells was about 10-fold less at 12,000 cells per well. We did not vary the number of tumor cells either. However, CAOV3 cells are smaller than OVCA429 and OVCA433 and they also proliferate very slowly. Therefore, there appear to be much fewer CAOV3 cells in each field. Similar to the mesothelial cells, the THP-1 cells seemed to be somewhat displaced by the presence of tumor cells. Yes, the left columns in 2B represent OCC morphology. We apologize for the typographical error.

  1. It is unclear how large is the response of OCCs to macrophages? What fraction of the OCCs contain tunneling nanotubes in the various conditions?

This is an interesting question. The fraction of OCCs that contain TnTs varies by condition and by density. However, with EGF induction we typically see between 5-10% of the cells acquire TnTs. This is similar with macrophage conditioned media (except the OVCA433 cells had slightly more TnTs with conditioned media). TnTs generally formed on the edges of epithelial sheets or in smaller clusters of cells that have undergone EMT. Cells in the center of the epithelial sheets seemed to be “protected” from TnT formation. We chose not to plate cells too sparsely and skew the percentages. However, we are now conducting studies on the impact of macrophages on cancer stem cells/ spheroid cultures. We used criteria from Dr. Cox and Dr. Lou’s labs to assess TnTs numbers[8,9].

Some of the images are of too low magnification to clearly see cell morphology. Also, some of the structures indicated to be nanotubes appear to be too thick to be “nanotubes”.

We enlarged some of the figures for clarity. We completely agree that there are many membrane extensions that are too large to be TnTs. As example we noted in Figure S8.   We made the images of the TnTs clearer to indicate the ones believed to be TnT-like. We made a distinct effort to quantitate thin projections that connect 2 cells. If there were not thin at the base or did not connect to another cell, they were not counted. As a side note, when we plate cancer stem cells on THP-1 cells we get projections not normally found in CSC, that are TnT-like (bridges connecting cells) but very thick. They also frequently contain mitochondria.

Some of the long structures that don’t connect two cells together could be migratory tails. Is the EGF or TFGα inducing cell migration/ scattering or breaking down cell-cell junctions?

Yes, it is well established and there are many publications including from our lab demonstrating the EGF induces EMT, invasion, and migration[10-12]. Therefore, we did not examine EMT markers as this is established. However, we can see a loss of cell-cell junctions and cortical actin normally found in OVCA433 and OVCA429 cell cultures (Figure 4). The point is well taken and it can be hard to discern true TnTs. But we did not count membrane extensions that are thick or not connecting cells as TnTs.

  1. EGF and TGF induce multiple signaling pathways leading to tunneling nanotube formation. Why was MAPK signaling examined?

We examined a number of pathways downstream of EGFR signaling including MAPK (PD184352, U0126, SCH772984, SL0101), PI3K (wortmannin, LY29004), GTPase activation (ML141), NO signaling (L-NAME), and actin polymerization (CD666, SMIFH2). The only inhibitors that prevented TnT formation were those in the MAPK signaling pathway.

While nanotube formation is quantified in Fig. 4B it is not clear what ave%TnT represent.

We apologize for the lack of clarity. We have updated the text. We quantitated the number of TnTs in a 20x field (a minimum of 3 fields per condition) and counted how many cells were in a field to determine the percentage of cells with TnTs. It is worth noting that some cells had multiple TnTs.

Does it represent number of cells with TnTs? The text said that 100-300 cells/field. Are the error bars the SD or SEM of independent experiments or fields?

Thank you for noting the lack of clarity. We have addressed this in the text. We conducted a minimum of 3 independent experiments for each set of inquiries. For each experiment we counted a minimum of 3 independent fields. The number of TnTs was averaged for the 3 independent fields and the error bars are the SEM between independent experiments.

It is not clear how MAPK signaling is regulating TnT formation? Is this through gene expression or some other signaling pathway?

This is a wonderful question and one that is still being investigated. What we found is that inhibition of MEK, ERK, or the ERK target RSK leads to loss of TnT-like projections. Our current hypothesis is that RSK targets proteins that lead to TnTs this may or may not include transcriptional activation and protein synthesis. However, given that it takes >12 hours to see EGFR activation result in TnT formation, some level of gene expression and protein synthesis are likely involved. RSK has been demonstrated to be involved in invasion and metastasis. RSK does this by regulating transcription, integrin activity, and actin remodeling[2].

How does PI 3-kinase and Akt signaling relate to MAPK activity?

PI3K and AKT are down stream of receptor tyrosine kinase (RTK) activation such as EGFR (in addition to MAPK being downstream of RTKs). We wanted to explore the role of different pathways that are activated by EGFR in TnTs. In the described studies, PI3K signaling does not appear to be involved in TnT formation in these cells. We are sure to clarify our rationale in the text. Our data suggest that MAPK signaling contributes to TnT formation but PI3K signaling does not.

  1. Control images of cells treated with CM with DMSO should be shown in figure 5 as a positive control.

We will add additional control images to Figure 6 (previously 5).

A reference should be included for the Stack study.

Dr. Stack’s publication was noted in text and was originally reference 22. With the addition of new references, Dr. Stack’s publication is now reference # 29.

  1. Torchiaro, E.; Lorenzato, A.; Olivero, M.; Valdembri, D.; Gagliardi, P.A.; Gai, M.; Erriquez, J.; Serini, G.; Di Renzo, M.F. Peritoneal and hematogenous metastases of ovarian cancer cells are both controlled by the p90RSK through a self-reinforcing cell autonomous mechanism. Oncotarget 2016, 7, 712-728, doi:10.18632/oncotarget.6412.
  2. Sulzmaier, F.J.; Ramos, J.W. RSK isoforms in cancer cell invasion and metastasis. Cancer research 2013, 73, 6099-6105, doi:10.1158/0008-5472.CAN-13-1087.
  3. Davidowitz, R.A.; Iwanicki, M.P.; Brugge, J.S. In vitro mesothelial clearance assay that models the early steps of ovarian cancer metastasis. Journal of visualized experiments : JoVE 2012, 10.3791/3888, doi:10.3791/3888.
  4. Iwanicki, M.P.; Davidowitz, R.A.; Ng, M.R.; Besser, A.; Muranen, T.; Merritt, M.; Danuser, G.; Ince, T.A.; Brugge, J.S. Ovarian cancer spheroids use myosin-generated force to clear the mesothelium. Cancer discovery 2011, 1, 144-157, doi:10.1158/2159-8274.CD-11-0010.
  5. Roy, L.; Bobbs, A.; Sattler, R.; Kurkewich, J.L.; Dausinas, P.B.; Nallathamby, P.; Cowden Dahl, K.D. CD133 Promotes Adhesion to the Ovarian Cancer Metastatic Niche. Cancer growth and metastasis 2018, 11, 1179064418767882, doi:10.1177/1179064418767882.
  6. Peters, P.N.; Schryver, E.M.; Lengyel, E.; Kenny, H. Modeling the Early Steps of Ovarian Cancer Dissemination in an Organotypic Culture of the Human Peritoneal Cavity. Journal of visualized experiments : JoVE 2015, 10.3791/53541, e53541, doi:10.3791/53541.
  7. Hu, J.; Zhang, N.A.; Wang, R.; Huang, F.; Li, G. Paclitaxel induces apoptosis and reduces proliferation by targeting epidermal growth factor receptor signaling pathway in oral cavity squamous cell carcinoma. Oncol Lett 2015, 10, 2378-2384, doi:10.3892/ol.2015.3499.
  8. Carter, K.P.; Segall, J.E.; Cox, D. Microscopic Methods for Analysis of Macrophage-Induced Tunneling Nanotubes. Methods in molecular biology 2020, 2108, 273-279, doi:10.1007/978-1-0716-0247-8_23.
  9. Desir, S.; O'Hare, P.; Vogel, R.I.; Sperduto, W.; Sarkari, A.; Dickson, E.L.; Wong, P.; Nelson, A.C.; Fong, Y.; Steer, C.J., et al. Chemotherapy-Induced Tunneling Nanotubes Mediate Intercellular Drug Efflux in Pancreatic Cancer. Scientific reports 2018, 8, 9484, doi:10.1038/s41598-018-27649-x.
  10. Cowden Dahl, K.D.; Dahl, R.; Kruichak, J.N.; Hudson, L.G. The epidermal growth factor receptor responsive miR-125a represses mesenchymal morphology in ovarian cancer cells. Neoplasia 2009, 11, 1208-1215.
  11. Cowden Dahl, K.D.; Symowicz, J.; Ning, Y.; Gutierrez, E.; Fishman, D.A.; Adley, B.P.; Stack, M.S.; Hudson, L.G. Matrix metalloproteinase 9 is a mediator of epidermal growth factor-dependent e-cadherin loss in ovarian carcinoma cells. Cancer research 2008, 68, 4606-4613.
  12. Cowden Dahl, K.D.; Zeineldin, R.; Hudson, L.G. PEA3 Is Necessary for Optimal Epidermal Growth Factor Receptor-Stimulated Matrix Metalloproteinase Expression and Invasion of Ovarian Tumor Cells. Molecular cancer research : MCR 2007, 5, 413-421.

Reviewer 2 Report

This is nice paper describing tunneling nanotubes formation between ovary cance cells and macrophages, which is a critical phenomenon facilitating metastasis. The paper is original and well done. However, the authors must better guide the reader by much more developed descriptions of figures. The figures definitly require more arrows and hints to follow all detils. A reader not familiar with these particular cell types is easily lost. The legends for figures should be much more devloped to facilitate reader`s analysis.

minor points:

l.95 cells expresed elevated expression...   redundant

Author Response

Thank you for the helpful review, reply is noted in bold font.

This is nice paper describing tunneling nanotubes formation between ovary cance cells and macrophages, which is a critical phenomenon facilitating metastasis. The paper is original and well done. However, the authors must better guide the reader by much more developed descriptions of figures. The figures definitly require more arrows and hints to follow all detils. A reader not familiar with these particular cell types is easily lost. The legends for figures should be much more devloped to facilitate reader`s analysis.

We greatly appreciate the constructive comments provided they have been taken into consideration and used to improve the manuscript. Thank you for noting the lack of clarity and description of figures. We have modified the paper extensively to provide more descriptions. Arrows were better placed. Cell types are better described. And more details are provided in the legends. These clarifications enhance the narrative of the manuscript.

minor points:

l.95 cells expresed elevated expression...   redundant

The manuscript has been extensively revised for typos.

Reviewer 3 Report

1) Background: More details on TNTs are needed.  There are multiple different kinds. 

2) Background: Results should not be included in the introduction. 

3) Results: Data presented in Lines 92 to 96 doesn't say what was measured (RNA or protein)  how it was measured (flow or qPCR) does not include the number of replicates, controls or how data analysis was performed.  it does not include the actual data generated or refer to a figure.  As such, conclusions drawn in line 96-97 are not supported by the data.  The rest of the manuscript suffers from the same flaws. 

4) Results: The conclusions that there are interactions between OCC and mesothelial and omental adipocytes us not supported by the figures.  Just because cells are co-cultured does not mean they are interacting.

5) Results: CD14 is a monocyte marker, and does not provide evidence that cells have been differentiated into macrophages. its not surprising that the results in line 162 show similar effects across all macrophage phenotypes. 

6) Results: Evidence of TNTs is not compelling.  No functional assays are included and the microscopy method is insufficiently described.  I'm not sure EVOS M5000 has the resolution to detect TNTs, most papers use a super resolution method

7) Methods: Lacking in details for many experiments.  Most glaring are the microscopy, analysis methods including for fig 4. 

Author Response

We thank the reviewer for the critical feedback that was used as a springboard to improve the data and text. The paper has been enhanced through these comments.

  • Background: More details on TNTs are needed.  There are multiple different kinds. 

A longer discussion of TnTs are provided in the introduction.

  • Background: Results should not be included in the introduction. 

The background has been edited to remove mention of results.

3) Results: Data presented in Lines 92 to 96 doesn't say what was measured (RNA or protein)  how it was measured (flow or qPCR) does not include the number of replicates, controls or how data analysis was performed.  it does not include the actual data generated or refer to a figure.  As such, conclusions drawn in line 96-97 are not supported by the data.  The rest of the manuscript suffers from the same flaws. 

Thank you for the constructive criticism.  These experiments were originally conducted by qRT-PCR array.  We have removed them for this draft of the paper in order to focus the macrophage results and more align the narrative.

4) Results: The conclusions that there are interactions between OCC and mesothelial and omental adipocytes us not supported by the figures.  Just because cells are co-cultured does not mean they are interacting.

Thank you for the critique.  We were trying to say that the OCCs maintain their epithelial morphology when cultured with mesothelial cells.  Additionally, when the OCCs are cultured with the omental adipocytes, they are largely epithelial but display some disorganization and membrane projections. We have adjusted the text to reflect this.

5) Results: CD14 is a monocyte marker, and does not provide evidence that cells have been differentiated into macrophages. its not surprising that the results in line 162 show similar effects across all macrophage phenotypes. 

We will clarify the text regarding the use of CD14.  Yes, CD14 is a monocytic marker.  CD14 was used in the immunofluorescence of the THP-1/OCC co-cultures to denote a difference between tumor cells and immune cells. We did not mean to use CD14 as a differentiation marker. The text has been corrected.  Our RT-PCR demonstrates that the cells treated with IFNg and LPS had increased mRNA expression of TNF which is consistent with an M1 phenotype.  The cells treated with IL13 and IL4 had increased mRNA expression of FN which is consistent with and M2 phenotype. Additionally, published reports demonstrate that THP-1 cells differentiated with PMA express macrophage the macrophage marker CD68[1].  We have cited this reference and added our immunofluorescence of CD68.

6) Results: Evidence of TNTs is not compelling.  No functional assays are included and the microscopy method is insufficiently described.  I'm not sure EVOS M5000 has the resolution to detect TNTs, most papers use a super resolution method

We understand the concern.  It is a valid concern when describing TnTs.  It would be an easier task if TnTs had a definitive marker. While we did not have access to super resolution imaging, we used published criteria form Dr. Lou and Dr. Cox’s labs to assess if actin-based membrane bridges were consistent with TnTs[2,3].  We will acknowledge the potential for the structures to not be TnTs in the text.

For supplemental data we added some higher magnification images including co-staining of the mitochondria protein Tom20 with phalloidin, which demonstrates that some of the TnT-like structures contain mitochondria.  We will adjust the language and augment the conclusion to address the concerns.

7) Methods: Lacking in details for many experiments.  Most glaring are the microscopy, analysis methods including for fig 4. 

The methods section has been extensively revised to include all necessary details for experimental  procedures and analysis including the microscopy section.

  1. Genin, M.; Clement, F.; Fattaccioli, A.; Raes, M.; Michiels, C. M1 and M2 macrophages derived from THP-1 cells differentially modulate the response of cancer cells to etoposide. BMC cancer 2015, 15, 577, doi:10.1186/s12885-015-1546-9.
  2. Carter, K.P.; Segall, J.E.; Cox, D. Microscopic Methods for Analysis of Macrophage-Induced Tunneling Nanotubes. Methods in molecular biology 2020, 2108, 273-279, doi:10.1007/978-1-0716-0247-8_23.
  3. Desir, S.; O'Hare, P.; Vogel, R.I.; Sperduto, W.; Sarkari, A.; Dickson, E.L.; Wong, P.; Nelson, A.C.; Fong, Y.; Steer, C.J., et al. Chemotherapy-Induced Tunneling Nanotubes Mediate Intercellular Drug Efflux in Pancreatic Cancer. Scientific reports 2018, 8, 9484, doi:10.1038/s41598-018-27649-x.

Round 2

Reviewer 1 Report

Overall, the revised manuscript is greatly improved with additional information and discussion.  There are still some additional issues that need to be resolved. Mainly, the TNTs are very hard to see in some of the images due to the low intensity of images and the low magnification. The induced structures may not indeed be true TnTs and may be more akin to tumor microtubes seen in glioblastoma.

Minor issues

1.       Line 82 – change cells to cell

2.       Line 119 – In figure 1E are the OVCA429-CD133-GFP alone or with adipocytes? What is the difference between 1E and 1F?

3.       Line 137 – 2.4.1 should be on a new line

4.       Line 150 – protrusions is misspelled

5.       Line 159 – “were differentiated and differentiated and polarized” Do you mean differentiated or differentiated and then polarized to M1 and M2?

6.       Line 166-169 – In Figure 3 EGF and TNF is not included in the figure. There are no panels labeled A-I and all images are of the same magnification.  Is this description part of Figure 4 legend?

7.       Line 238 – sufficient is misspelled

8.       The TnTs in the control panels (THP-1 M0 + DMSO) in Figure 6 cannot be seen.

9.       Line 290 – remove period at the beginning of the sentence

10.   Please clarify when is meant by the sentence in lines 293-294. It is not clear from the images in Figure 1 that the OCCs did not clear the adipocytes but were “seeded” or remained on top.

Author Response

Dear Reviewer 1:

We thank you for your careful critique of our manuscript as it has improved the quality of the paper. We have made the requested. We acknowledge that some of the structures may not be TnTs and that there are other possibilities for membrane structures that may be induced. We did state this, however, we will emphasize it further.

  1. Line 82 – change cells to cell

The change has been made.

  1. Line 119 – In figure 1E are the OVCA429-CD133-GFP alone or with adipocytes? What is the difference between 1E and 1F?

We apologize for the confusion. As stated in the text and figure legend, Figure 1E and 1F are OVCA/adipocyte co-cultures. Figure 1E is with the OVCA429 cells (OVCA429-CD133-GFP) and Figure 1F is with the OVCA433 cell line (OVCA433-CD133-GFP). Panels A and B are the only cells in this figure not co-cultured. We have added text to make this clearer.

  1. Line 137 – 2.4.1 should be on a new line

It is unclear what to do with this comment. We do not have a 2.4.1. But we do have a 2.3.1. We have check spacing to eliminate any inconsistencies.

  1. Line 150 – protrusions is misspelled

The correction was made.

  1. Line 159 – “were differentiated and differentiated and polarized” Do you mean differentiated or differentiated and then polarized to M1 and M2?

Yes, we mean that some cells were differentiated, and some were differentiated and then subsequently polarized to M1 or M2. We have clarified this in the text.

  1. Line 166-169 – In Figure 3 EGF and TNF is not included in the figure. There are no panels labeled A-I and all images are of the same magnification.  Is this description part of Figure 4 legend?

We apologize for any confusion. Line 166-169 where Figure 3 is discussed is in the first submission of the manuscript and not the revision. We have double checked and Figure 3 and 4 are correctly cited in the current version.

  1. Line 238 – sufficient is misspelled

      The correction has been made.

  1. The TnTs in the control panels (THP-1 M0 + DMSO) in Figure 6 cannot be seen.

We apologize and have supplied new images that are comparable with those in Figure 3.

  1. Line 290 – remove period at the beginning of the sentence

Thank you we have edited for typos.

  1. Please clarify when is meant by the sentence in lines 293-294. It is not clear from the images in Figure 1 that the OCCs did not clear the adipocytes but were “seeded” or remained on top.

I apologize. This was not a major point. We were emphasizing that (as it has been previously shown) mesothelial cells are displaced by OCCs. The adipocytes were not similarly displaced. This is most clearly seen in Figure 1F where GFP-OCCs can be seen on top of green lipid droplets. The text has been edited for clarity.

Reviewer 3 Report

1) I find no mention of the number of replicates (with exception of figure 5) in either the results or the methods. For example, how many replicates of macrophage repolarization were performed?

2) I can not open the Supplemental data file, so I am unable to view the results of the RNA or Westerns, etc.

3) Section 2.5- how was MAPK signaling measured or evaluated? TNT formation is not proof of MAPK activation.

4) The arguement for TNT formation remains unconvincing

5) The field selected for evaluating TNTs appears highly subjective (lines 212-220).  Suggesting a decrease from 7-10% to 2-3% is a drastic reduction is also a stretch.

6) MAPK signaling pathway includes a huge number of factors.  U0126 specifically inhibits ERK1/2 (not MEK), so the pathway that is being inhibited theres RAS/ RAF/MEK and not TRAF and RAC1 pathways

6) Lines 500-505, the conclusion that MAPK signaling is initiated is not supported by the data

Author Response

Dear Reviewer 3:

We thank you for your careful critique of our manuscript as it has improved the quality of the paper. We have made the requested changes. We acknowledge that some of the structures may not be TnTs and that there are other possibilities for membrane structures that may be induced. We did state this, however, we will emphasize it further. We appreciate the critiques that serve to strengthen the paper.

  1. I find no mention of the number of replicates (with exception of figure 5) in either the results or the methods. For example, how many replicates of macrophage repolarization were performed?

We apologize for this oversight that has now been corrected. All experiments were performed a minimum of 3 times. The only exception to this is that while we did perform multiple independent replicates of the OCC/adipocyte co-cultures, due to the expense of the adipocytes we only used 2 independent adipocyte cell lines. Text has been amended to state the replicates for each set of experiments.

  1. I can not open the Supplemental data file, so I am unable to view the results of the RNA or Westerns, etc.

I apologize for the technical difficulties.

  1. Section 2.5- how was MAPK signaling measured or evaluated? TNT formation is not proof of MAPK activation.

Thank you for the question. We did not mean to imply that TnT formation is synonymous with MAPK activation. The GFP staining in Figure 7 comes from the myr-AKT-GFP fusion and the activated KRAS-GFP fusion. The cells being green indicate that the oncogenes are present in the cells. We conducted western blot for activated phosphorylated ERK1/2 on lysates form the KRAS expressing cells. Unfortunately, the blot was not conclusive for both the OVCA4429 and OVCA433 cells. The cells were not serum starved or treated with growth factors. Given that we only had 10 days to respond (including a weekend and the Christmas holiday) we failed to repeat the experiment. In order to not over analyze our results, we took out Figure 7. Importantly this does not change the conclusion of the paper that MAPK signaling is an important pathway leading to TnT formation.

  1. The arguement for TNT formation remains unconvincing

We have indicated throughout the manuscript that some of the structures may be TnT-like or other structures particularly in the discussion. We will soften the tone of the paper and emphasize that the main conclusion is that these structures TnT or other protrusions are a direct result of MAPK signaling.

  1. The field selected for evaluating TNTs appears highly subjective (lines 212-220).  Suggesting a decrease from 7-10% to 2-3% is a drastic reduction is also a stretch.

We agree that choosing a field of view to evaluate could greatly sway the results. Therefore, we intentionally used denser fields where there were both areas of epithelial sheets and more spread out cells. There were fields with far more TnT-like structures than we used for images or counted. This is why we chose to count/use images with around 200 cells stay consistent and to acknowledge that not all cells induced these structures in response to CM or EGF. If we included areas with more sparse cells that percentage of cells with TnTs would be much higher. The images are representative of our findings. We see virtually no membrane extensions in the absence of macrophage CM or EGFR activation. But we see extensive membrane projections like TnTs formed with CM or EGFR activation, which is lost with MAPK inhibition. The word drastically was removed. The reduction in TnTs was however, significant.

  1. MAPK signaling pathway includes a huge number of factors.  U0126 specifically inhibits ERK1/2 (not MEK), so the pathway that is being inhibited theres RAS/ RAF/MEK and not TRAF and RAC1 pathways

Thank you for your concerns. First, U0126 is a specific inhibitor for MEK1 and MEK2 as demonstrated in Favata et al (PMID9660836) in JBC in 1998. The paper (and many others) demonstrate that U0126 inhibits the kinase activity of MEK1 and MEK2 but not Raf or ERK. ERK activity is inhibited by U0126 due to its effects on MEK activity. However, U0126 is selective for MEK. However, your point is well taken. The MAPK signaling pathway leads to extensive activation of downstream signaling events and cellular outcomes. More experiments are needed to identify all of the events downstream of MEK/ERK/RSK that lead to TnT-like protrusions.

  1. Lines 500-505, the conclusion that MAPK signaling is initiated is not supported by the data

The conclusion has been changed to say that macrophages induce formation of TnT-like structures through MAPK signaling.